# Study on the Flower Induction Mechanism of *Hydrangea macrophylla*

**DOI:** 10.3390/ijms24097691

**Published:** 2023-04-22

**Authors:** Yun Liu, Tong Lyu, Yingmin Lyu

**Affiliations:** 1Beijing Key Laboratory of Ornamental Plants Germplasm Innovation & Molecular Breeding, China National Engineering Research Center for Floriculture, College of Landscape Architecture, Beijing Forestry University, Beijing 100083, China; 2Beijing Flower Engineering Technology Research Center, Plant Institute, China National Botanical Garden North Park, Beijing 100093, China

**Keywords:** *Hydrangea macrophylla*, flower induction, continuous flowering, flowering regulation pathway

## Abstract

The flower induction of *Hydrangea macrophylla* “Endless Summer” is regulated by a complex gene network that involves multiple signaling pathways to ensure continuous flowering throughout the growing season, but the molecular determinants of flower induction are not yet clear. In this study, genes potentially involved in signaling pathway mediating the regulatory mechanism of flower induction were identified through the transcriptomic profiles, and a hypothetical model for this regulatory mechanism was obtained by an analysis of the available transcriptomic data, suggesting that sugar-, hormone-, and flowering-related genes participated in the flower induction process of *H. macrophylla* “Endless Summer”. The expression profiles of the genes involved in the biosynthesis and metabolism of sugar showed that the beta-amylase gene *BAM1* displayed a high expression level at the BS2 stage and implied the hydrolysis of starch. It may be a signaling molecule that promotes the transition from vegetative growth to reproductive growth in *H. macrophylla* “Endless Summer”. Complex hormone regulatory networks involved in abscisic acid (ABA), auxin (IAA), zeatin nucleoside (ZR), and gibberellin (GA) also induced flower formation in *H. macrophylla*. ABA participated in flower induction by regulating flowering genes. The high content of IAA and the high expression level of the auxin influx carrier gene *LAX5* at the BS2 stage suggested that the flow of auxin between sources and sinks in *H. macrophylla* is involved in the regulation of floral induction as a signal. In addition, flowering-related genes were mainly involved in the photoperiodic pathway, the aging pathway, and the gibberellin pathway. As a result, multiple pathways, including the photoperiodic pathway, the aging pathway, and the gibberellin pathway, which were mainly mediated by crosstalk between sugar and hormone signals, regulated the molecular network involved in flower induction in *H. macrophylla* “Endless Summer”.

## 1. Introduction

In higher plants, the transition from the vegetative to reproductive stage occurs during flower induction. Morphogenesis is a complex process that is affected by genes, the developmental stage, physiological signals and external conditions [1]. In *Arabidopsis*, flowering is regulated by multiple pathways, such as the photoperiodic pathway, the vernalization pathway, the ambient temperature pathway, the gibberellin pathway, the autonomous pathway, and the aging pathway [2]. It has been reported that hormones and sugar induce flower formation processes [3]. Flowering is a very important trait in ornamental plants which determines their ornamental value. However, the regulatory mechanism of flower induction in *H. macrophylla* “Endless Summer” is still unclear. Studies have shown that flowering and development in plants are relatively genetically conserved. This study explored the molecular mechanism of flower induction in *H. macrophylla* “Endless Summer”, which is significant for understanding the flowering mechanism of woody plants, with the model plant *Arabidopsis thaliana* as a reference.

In the photoperiodic pathway, leaves respond to changes in day length through specific photoreceptors and the central oscillator [4]. The central oscillator is composed of a negative feedback loop, including *CCA1*, *TOC1*, *LHY*, *ELF4*, *PRR9*, and *LUX* [5]. Different oscillatory rhythms are generated, based on changes in the external photoperiod. The circadian expression of the output genes *GI* and *CO* converts the photoperiod signal into a flowering signal, ultimately inducing flowering in plants [6,7]. In the gibberellin pathway, GA signals can inhibit the expression of *SVP* genes, thereby relieving the inhibition of *FT* genes in the photoperiodic pathway [8]. On the other hand, GA promotes flowering by stimulating the transcription and expression of the floral tissue-determining gene *LFY* [9]. The aging pathway mainly relies on miR156 and miR172 to negatively regulate their target transcription factor genes and play a role in the transition from vegetative growth to reproductive growth [10]. The target genes of miR156 are members of the *SPL* gene family, which mainly exist as flowering promoting factors and can promote the expression of flowering integrators [11,12]. The target genes for miR172 are members of the *AP2* transcription factor family, which are inhibitory factors of flowering that can delay flowering [13]. Moreover, studies have shown that the aging pathway can induce flower bud differentiation independently of other flower induction pathways [14]. Flowering signals from various pathways are integrated into the flowering integrators *FT*, *LFY*, and *SOC1*, affecting the expression of the MADS-box transcription factor genes *AP1* and *FUL*, thereby regulating floral transformation and floral organ development [15].

Under specific circumstances, the control of hormone signals is frequently accomplished by alterations in the expression level of important flowering genes after the accumulation of various hormone signals. Gibberellin (GA), the primary signaling molecule in the gibberellin pathway, plays a crucial role in the blooming process, and other hormones, including IAA, CTK, and ABA, are also essential components of the hormone control network. To finish the flowering process, genes from several pathways converge on the flowering integron and support the development of floral organs [16].

*H. macrophylla* “Endless Summer” was used as the experimental material in order to explore the mechanism of the flower induction pathways. Phenotypic observations and determination of the endogenous hormones were implemented, and candidate genes related to flowering were screened using transcriptome sequencing.

## 2. Results

### 2.1. Phenotypic Observations during Flowering

Due to the cold temperatures in Beijing, the previously formed shoots and buds of *H. macrophylla* “Endless Summer” were destroyed. Basal shoots sprouted from the plant’s base in the spring once the temperature rose. In this study, which began on 20 March 2021, morphological observations of the shoot apical meristem (SAM) of several differentiation phases of basal shoots were carried out to clarify the flower development process. As seen in Figure 1a, during the vegetative period, the top of the growth point was flat, the outside contour was spindle-shaped, and then the stem tip started to expand, producing a dome in the middle (Figure 1b). Then it moved into the stage of meristem differentiation, going from three meristems (Figure 1c) to five meristems, and to nine meristems. The margin of the growing cone gradually broadened with the expansion of the meristem, and the differentiation of the floral primordium started (Figure 1d). The buds progressively grew larger and transitioned from a spindle form to a spherical shape (Figure 1e). The differentiation period was short, and there was no clear distinction in time between the aforementioned types. Following that, there were primary inflorescence branches (Figure 1f). Flower induction was completed and went into the stage of floral organ development when secondary inflorescence branches formed (Figure 1g). In Figure 1h, a few ornamental flower sepals developed first. As the sepals gradually expanded, the color changed to varying degrees (Figure 1i,j). The non-decorative blooms in Figure 1k were still green, but the sepals of the decorative flowers had fully opened and changed color. Both decorative and non-decorative flowers changed color and had partly unfolded petals at the peak flowering stage, as seen in Figure 1l.

### 2.2. Endogenous Hormone Response during Flowering

The contents of endogenous hormones such as ABA, IAA, GA_3_, and ZR in the seven development phases of BS1–BS7 were measured in order to examine the role of endogenous hormones in the flowering process of *H. macrophylla* “Endless Summer”. The levels of hormone content fluctuated significantly over the whole flowering process, as seen in Figure 2. After achieving the highest level in BS2, the concentration of ABA gradually fell, while BS4 had the lowest content: only 12.4% of the highest level. When the decorative flower sepals started to change color, the ABA content rose; following the complete color change, it subsequently significantly declined. The concentration of ABA present at full bloom was nearly identical to that in vegetative buds. At BS2 and BS3, the concentration of GA_3_ was lower before rising quickly. The maximum concentration of GA_3_ was found at BS4, which was 1.9 times higher than BS2’s lowest point. While the sepals of decorative flowers were becoming colored, the content of GA_3_ continued to drop, but as the flowers began to bloom, it began to rise. ZR was not highly expressed throughout flower induction from BS1 to BS4, and its concentration was lowest at BS3, rising quickly from BS4 to BS5. At the stage of blooming, the concentration of ZR was slightly lower than it had been during the coloring of decorative flowers.

### 2.3. Quality Assessment of Transcriptome Sequencing

By comparing the transcriptomes of *H. macrophylla* “Endless Summer” at seven different developmental stages, the cDNA library was created. Among them, BS1 and BS2 represented mixed samples of vegetative and induced buds, respectively, while BS3 through BS7 represented mixed samples of decorative flowers gathered during the growth season. In total, 49.2 G of clean data were produced in this investigation. The effective data volume of each sample was 6.79–7.36 G, the Q30 base distribution ranged from 88.66% to 91.86%, and the average GC content was 45.48% (Table 1). These results showed that the transcriptome sequencing was of a high standard and satisfied the criteria for subsequent data processing.

The statistical analysis of the genes expressed in seven samples yielded a total of 47,793 unigenes. The N50 length was 1785 bp, while the average length was 1177.32 bp (Table 2). This implied that the high assembly integrity of the transcriptome sequencing data satisfied the standards for data analysis.

### 2.4. Analysis of KEGG Metabolic Pathways and GO Functional Classifications

To understand the specific functions of the unigenes obtained by transcriptome sequencing during the development of *H. macrophylla* “Endless Summer” and the metabolic pathways the genes are involved in, the unigenes were mapped to the KEGG database. The statistical analysis resulted in a total of 5583 unigenes that were involved in 133 metabolic pathways. DEGs were enriched in carbohydrate metabolism and plant hormone signal transduction pathways (Table 3). The GO enrichment analysis resulted in flower-related GO terms (Table 4), such as flower development (GO:0009908), circadian rhythm (GO:0007623), and pollen development (GO:0009555).

### 2.5. Expression Profiles of Sugar-Related Genes during Flower Development

Genes representing carbohydrate metabolism and sugar transport during flower development were clustered, as shown in Figure 3. A hierarchical cluster analysis grouped these genes into seven major clusters. Cluster 1 included 20 genes that had increased expression levels from BS6 to BS7 and were mainly related to the interconversion of pentose and glucuronate, as well as sucrose synthase, such as *SUS1*. Genes in Cluster 2 (26 genes) typically showed low expression in the BS1–BS4 stage and then increased gradually until the BS7 stage. Among them, the pectate lyase 5 and pectinesterase 41 genes were involved in pentose and glucuronate interconversions, and the glyceraldehyde 3-phosphate dehydrogenase gene was involved in glycolysis. In addition, the sugar transporter gene *ESL1* and the bidirectional sugar transporter gene *SWEET1* also exhibited this expression profile. The expression profiles of nine genes in Cluster 3 typically displayed a sharp decrease between the BS1 and BS2 stages, and then remained constant or slightly declined during the BS2–BS7 stages. These genes were involved in the metabolism of amino sugar and nucleotide sugar, such as *CHI5* and *CHLY*, and the metabolism of starch and sucrose, such as *AMYB*. The expression profiles of 14 genes in Cluster 4 typically displayed an increase between the BS1 and BS2 stages, and then declined between the BS2 and BS3 stages and remained constant thereafter. These genes were involved in the metabolism of starch and sucrose, such as beta-amylase 1 (*BAM1*) and trehalose 6-phosphate phosphatase (*TPPJ, TPPA*), and in pentose and glucuronate interconversions, the metabolism of glyoxylate and dicarboxylate and so on. Interestingly, the genes in Cluster 5 (31 genes) displayed the completely opposite expression profile compared with the genes in Cluster 2, which showed high expression during BS1–BS4. The genes were involved in the metabolism of starch and sucrose, such as glucose-1-phosphate adenylyltransferase (*GLGL2*) and endoglucanase 24 (*GUN24*), and in the metabolism of amino sugars and nucleotide sugars. Genes in Cluster 6 (30 genes) displayed low expression during BS1–BS4, then increased until BS6 and decreased between BS and BS7. Among them, the pectate lyase genes *PLY5*/*8*/*11*/*22* and the pectinesterase gene *PME* were involved in pentose and glucuronate interconversions, as well as *GUN6*/*17*, which was involved in the metabolism of starch and sucrose. Cluster 7 included 12 genes that had increased in expression levels between BS3 and BS4 and were involved in multiple processes, such as pentose and glucuronate interconversions, glycolysis and the metabolism of amino sugars and nucleotide sugars, and so on. In general, sugar plays an important role in the entire flower development process.

### 2.6. Expression Profiles of Hormone-Regulatory Genes during Flower Development

Genes encoding plant hormone signal transduction pathways were analyzed and showed a significant change in their expression levels (Figure 4) of at least twofold among the different stages of flower development. Six main gene clusters were determined by hierarchical clustering based on their expression profiles during flower development. Genes in Cluster 1 (44 genes) typically displayed low expression during the BS1–BS4 stages and then increased gradually until the BS7 stage. The genes that respond to auxin, such as *SAU20*, *SAU24*, *SAU50*, and *IAA14*; auxin transport-related genes, such as *LAX2*; and auxin-activated signaling pathway genes, such as *AX22D*, *AX10A*, and *A10A5*, belonged to this cluster. In addition, the gene that responds to cytokinin (*ARR-A*) and the key gene related to ABA synthesis (nine-cis-epoxycarotenoid dioxygenase (*NCED*)) had the same expression profiles. The auxin-responsive protein genes *SAUR36* and *SAUR32*, and the gibberellin receptor gene *GID1B* were the members of Cluster 2 (four genes). The cytokinin-activated signaling pathway genes *AHP* and *ARR-A*, and the abscisic acid receptor gene *PYL* belonged to the third cluster (eight genes). The expression profiles of nine genes in Cluster 4 displayed a sharp increase between BS3 and BS4, decreased between BS4 and BS5, and then remained constant during other stages. The genes were mainly the abscisic acid receptor genes *PYL3*, *PYL4*, and *PYL8*, and the indole-3-acetic acid-amido synthetase gene *GH3*. Genes within Cluster 5 (14 genes) peaked at the BS2 stage and maintained low expression levels during the other stages. Genes associated with auxin-response, such as *SAU36* and *ARFA*; auxin transport, such as *LAX5*; and the reception of abscisic acid, such as *PYL4*, appeared in this cluster. In addition, DELLA protein genes and the two-component response regulator gene *ORR4* also grouped in this cluster. Interestingly, the genes in Cluster 6 (31 genes) displayed the complete opposite expression profile compared with the genes in Cluster 1, which showed high expression during BS1–BS4. The genes that respond to auxin, such as *IAA32* and *ARFE*, and auxin transport-related genes, such as *LAX1*, belonged to this cluster. In addition, the abscisic acid receptor gene *PYL8* and the gibberellin receptor gene *GID1B* were present in this cluster.

### 2.7. Expression Profiles of Flowering Genes during Flower Development

The expression profiles of flowering genes during flower development were analyzed by hierarchical clustering, which grouped these genes into five clusters (Figure 5). Cluster 1 included 12 genes that decreased in expression from the BS1 to BS2 stages, then remained constant or slightly decreased during BS2–BS7. Genes associated with the vegetative to reproductive phase transition of the meristem, such as *TPS1* and *TCP3*; axial regulation, such as *YAB5*; and *SPL4* and *SPL13* of the squamosa-promoter binding protein-like gene family, displayed this expression profile. MADS-box family members genes, such as *AGL24* and *SVP*; the early flowering gene *ELF4*; and the phytochrome gene *PHYB*, which regulates flower development, were grouped in this cluster. Genes within Cluster 2 (10 genes) peaked at the BS2 stage and maintained low expression levels during other stages. These genes were involved in the metabolism of starch and sucrose, such as trehalose 6-phosphate phosphatase (*TPPA* and *TPPJ*) and beta-amylase (*BAM1*). In addition, genes associated with the vegetative to reproductive phase transition of the meristem, such as *FLP2*, as well as *TI10A* and *PAT1*, which are involved in flower development, also displayed in this profile. Genes in Cluster 3 (eight genes) typically displayed high expression levels during the BS1–BS2 stages and then decreased gradually until the BS7 stage. MADS-box transcription factor family protein genes associated with flower development and floral meristem determinacy, such as *SVP* and *SOC1*; flower development and regulation genes, such as *FD*; and *SPL7* and *SPL17*, which are members of the SBP family, exhibited this expression pattern. The genes in Cluster 4 (nine genes) displayed high expression levels during the BS2–BS4 stages and low expression during the other stages. These genes were mainly involved in flower development, such as *FLO* and *UFO*, and in floral organ formation, such as *AP1*, *AP2*, and *FUL*. In addition, a circadian rhythm-related gene (*PRR5*) and a squamosa promoter-binding-like protein gene (*SPL3*) also exhibited this expression pattern. Cluster 5 included 19 genes, some that increased in expression during the whole process of flower development and some that displayed low expression levels during BS2–BS4 and high expression during BS5–BS7. Genes associated with floral organ development, such as *AG*, *EJ2*, and *TM6*, and with cell differentiation, such as *TCP2*/*4*/*13* were grouped within this cluster. In addition, circadian rhythm-related genes, such as *CRY1*, *RVE8* and *LHY*; flower development genes, such as *CDF2/3*; and the centrally regulated genes of the photoperiod pathway, such as *GI* and *COL9*, also exhibited this expression pattern.

### 2.8. Co-Expression Pattern Analysis

By performing a correlation analysis of the genes associated with plant hormone signal transduction, synthesis and transport of sugars, and flowering, a co-expression network with 158 nodes and 1586 edges was generated (shown in Figure 6). The edges between the nodes indicated a gene interaction. In contrast to tiny yellow nodes, which indicated that the gene was weakly connected, huge dark green nodes indicated that the gene was strongly connected. The auxin transporter-like protein gene *LAX5*; the auxin response genes *ARFE*, *SAU24*, *SAU20*, and *IAA14*; the auxin-inducible protein genes *A10A5* and *AX10A*; the auxin efflux carrier protein genes *PIN6*, *PIN3*, and *PIN1B*; and *GBSS*, *GLGA*, and *BAM1*, which are involved in the metabolism of starch and sucrose, are some of those with strong linkage. Five hub genes were identified using Cytoscape’s Cytohubba plug-in, including *BAM1*, *IAA14*, *G2OX6*, *LAX5*, and *GLGA*. The high expression levels of *BAM1* and *IAA14* in BS2–BS4 suggested that they playean important role in the flower induction, whereas the expression of *G2OX6*, *LAX5*, and *GLGA* in BS5–BS7 indicated that they were involved in the flowering period.

### 2.9. Real-Time Quantitative PCR Verification of Differentially Expressed Genes

Six genes related to flower induction in *H. macrophylla* “Endless Summer” were selected, namely *AP1*, *FT, LFY*, *COL9*, *GAOX2*, and *SPL9*, and their expression patterns were verified by qRT-PCR, as shown in Figure 7. The reliability of the transcriptome data was demonstrated by the correlation coefficient between their expression levels and transcriptome data, which was higher than 0.9.

## 3. Discussion

### 3.1. Sugar Signaling Mediates Flower Induction in H. macrophylla “Endless Summer”

Studies have shown that carbohydrates play an important role in plant growth and the flowering process. Sugar is not only an important energy source in the process of flower buds’ differentiation, but is also a key floral signal that initiates flower induction [17]. However, limited information is available on the specific effects of sugars on flower induction in perennial woody plants, especially *H. macrophylla*. Our results showed that the gene expression levels of the sucrose synthase gene *SUS1*; the amylase genes *BAM1* and *AMY3*; and the soluble starch synthase gene *GLGA* in the SAM of *H. macrophylla* “Endless Summer” significantly changed during the flower development stage, which may be due to their involvement as an energy source in the regulation of flower induction. The high expression of the amylase gene *BAM1* (one of the hub genes) at the BS2 stage implied the hydrolysis of starch, and it may be a signaling molecule that promotes the transition from vegetative growth to reproductive growth in *H. macrophylla* “Endless Summer”. Similar results were reported in *Arabidopsis thaliana* by Matsoukas [18]. In *Arabidopsis*, *HEXOKINASE 1* (*AtHXK1*) is a glucose receptor that integrates signal transduction pathways such as nutrients and hormones, and plays a crucial role in plant growth and development [19]. Different sugar signals are sensed and transduced through *HXK1*. In this study, *HXK1* was highly expressed in BS2, suggesting a high sugar level in the SAM during flower bud differentiation, which promoted flower induction. It has been reported that there is a significant correlation between the expression of flowering genes and sugar-related genes [20]. Currently, it is generally accepted that the signal molecule trehalose-6-phosphate (T6P) is crucial for the regulation of flowering and operates as a proxy for carbohydrate status in plants [21]. The expression of *TPPA* and *TPPJ* was high in the BS1–BS2 stages, while the sucrose synthase gene *SUS1* was highly expressed in the BS5–BS7 stages, indicating that T6P and sucrose are both indicators of plant carbohydrate status. Some studies have shown that sucrose and T6P levels are tightly associated, and elevating the expression of sucrose in plants can also raise the levels of T6P, which can affect the transduction and metabolism of sugar signals and initiate flower induction [22]. In addition, sucrose can promote the positive expression of *GI*, thereby participating in floral induction [23]. The expression levels of *GI* in the transcriptome data were higher during the BS5–BS7 stages. In addition, some members of the SBP transcription factor gene family, such as *SPL7* and *SPL9*, as miR156 target genes involved in the aging process, showed high expression levels in BS2 and decreased with bud growth, similar to the situation observed in Arabidopsis [24]. This suggests that the sugars involved in the induction of flowering in *H. macrophylla* are involved in the aging pathway of flower induction, as shown in Figure 8.

### 3.2. Hormone Signal-Mediated Flower Induction in H. macrophylla “Endless Summer”

As an important endogenous signal, plant hormones play an important role in the flowering process. Endogenous plant hormones can regulate plants’ growth and development by binding to specific protein receptors and regulate the process of bud differentiation in woody plants [25]. In this study, we identified the hormone levels during flower development and the expression levels of genes associated with hormone regulation and responses in order to understand the role of hormones in flower induction and flowering. Abscisic acid is an important hormone that plays a key regulatory role in different stages of plant development and in plants’ responses to environmental stresses [26]. Our results showed that abscisic acid exhibited a high expression level and reached the highest expression level at the BS2 stage during the induction of flowering. At the end of flower induction, the ABA content decreased, and then rose again at the blooming stage. According to our results, the expression patterns of *NECD* (involved in ABA biosynthesis) and of homologous ABA signal transduction genes, such as *PYL* and *SNRK2*, displayed similar changes to the ABA content in the SAM, showing high expression levels at the BS2 stage of flower induction, and then another increase at the flowering stage. This indicated that this change may mediate the initiation of flower formation and the regulation of the flowering process in *H. macrophylla*, shown in Figure 8. In *Arabidopsis thaliana*, the activation of flower induction requires the photoperiodic response gene *GI* and the florigen gene *FT*, and the ABA-dependent activation of *FT* requires *CO* for promoting the transcriptional upregulation of the florigen genes [27]. In addition, studies have shown that ABA induces flower induction by promoting the expression level of *AP1* [28]. In our study, the change trend of the expression levels of *AP1* was similar to that of ABA. Our results shows that ABA participated in the regulation of flower induction in *H. macrophylla* “Endless Summer” through various pathways.

Cytokinin controls many aspects of growth and development in woody plants. Zeatin nucleoside is the main type of CTK transport in the xylem, and is important for the differentiation of flower buds [29]. In fact, our results showed that the expression level of ZR in SAM continued to decrease from BS1 to BS3 until the flowering induction process was completed, and significantly increased during the flowering process. The key genes of ZR synthesis, *ZOX* and *ZOG*, also showed the same change trend, while the key genes for the CTK signal transduction pathway, such as *ARR* and *AHP*, were highly expressed in the BS1–BS2 stages. This indicated that ZR may participate in the flower induction process as a signal molecule during the first stage of flower induction. Studies have shown that *AUXIN RESPONSE FACTOR* (*ARF*) integrates the activities of *AGAMOUS* (*AG*) and auxin to control the biosynthesis and signaling of cytokinin, thereby coordinately regulating the maintenance and termination of the floral meristem [30]. In our study, the expression of *ARF* was high at the BS2 stage while the content of zeatin nucleoside decreased from the BS1 to BS3 stages, suggesting that *ARF* regulates the floral meristem’s determinacy by repressing the biosynthesis and signaling of cytokinin. In addition, it was reported that ZR can downregulate the expression level of miR172 and promote the expression of *AP2* protein (shown in Figure 8), thereby promoting the regulation of the flowering process of *H. macrophylla* “Endless Summer” [31].

Gibberellin plays a role in flower induction in model plants such as *Arabidopsis* through the gibberellin pathway [32]. Our results showed that the expression of gibberellin in flower induction was low, and the expression of gibberellin in the flowering stage was high, and the key gene for gibberellin biosynthesis *G2OX6* had the same expression trend. In addition, GAs can activate the expression of *LFY* through cis-acting elements that are different from those that respond to daylength [33], indicating that the environmental and intrinsic signals that control flowering can be integrated on the *LFY* promoter.

Auxin affects the elongation, differentiation, and floral induction processes of plants, and was the earliest identified plant hormone [34]. Our results exhibited a high expression level of auxin in the SAM during flower induction, while the differential expression of genes related to the auxin biosynthesis process was not significant. At the same time, the auxin influx carrier gene *LAX5* was highly expressed during flower induction, while *LAX2* was highly expressed during the flowering stage. This indicated that the flow of auxin between sources and sinks in *H. macrophylla* acts as a signal, which was involved in the regulation of flower induction. Kiyotaka’s research showed that the normal level of the auxin polar transport system is necessary for the development of inflorescences and flower buds [35]. In this study, the expression of the auxin polar transport gene *PIN6* was high at the BS2 stage, indicating that it provided conditions for the development of inflorescences and flower buds in *H. macrophylla*. Flower primordia arose in the periphery of the inflorescence meristem at the sites of auxin maxima during the reproductive stage in *Arabidopsis thaliana.* At these sites, *ARF5*/*MP* upregulated the expression of *LFY*, via evolutionarily conserved and biologically important cis-regulatory motifs in the *LFY* promoter to specify these primordia as flowers and promote their outgrowth (shown in Figure 8) [36]. In addition, IAA can also participate in the processes of GA biosynthesis and signal transduction by promoting the expression of *G2OXs* and inhibiting the expression of DELLA protein [37].

### 3.3. Flower Induction Pathway in H. macrophylla “Endless Summer”

Several environmental signals, such as daylength (photoperiod), winter (vernalization), high ambient temperatures, and endogenous signals, including plant age (aging), the plant hormone GA, and other factors all influence flower induction in the model plant *Arabidopsis thaliana* [38]. Flowering signals from different flower induction pathways are integrated into the flower integrator, such as *FT*, *LFY*, and *SOC1*, thereby regulating the determination of the floral meristem and the development of floral organs, and completing the regulation of flower induction [39]. The hypothetical model of a complex gene network of the regulatory mechanism of sugars, hormones, and multiple pathways, including the photoperiodic pathway, the gibberellin pathway, and the aging pathway, which mediate and regulate flower induction in *H. macrophylla* “Endless Summer”, can be seen in Figure 8. In this study, we used clustering methods and analyzed the expression profiles of flowering genes involved in the flower induction pathway during the flowers’ development (Figure 5). Our results showed that *ELF4*, *PRR5*, *LHY*, and *RVE8* participated in the circadian rhythm. The photosensitive photoreceptor pigment *PHYB* and the cryptochrome promoting flowering *CRY1* sense the photoperiod by absorbing red and far red light, thereby transmitting signals to the circadian clock [40]. *GI* is an output gene of the circadian clock. In our study, the expression of *GI* continuously increased to promote flowering. The circadian clock further transmitted signals of photoperiodic changes to *COL9* through *CDFs*, while *PHYB* and *CRY1* also promoted the expression of *COL9*. *COL9* transmitted signals to *FT*, which upregulated the expression level of *FT* to control the transition to flowering in the photoperiodic pathway [41]. In addition, *FD*, a *bZIP* transcription factor interacted with *FT*, forming a complex that activated *AP1*, completing flower induction [42]. In our study, the expression levels of MADS family members such as *AGL24*, *SOC1*, etc., were higher at the early stages of flower induction and then gradually decreased. Studies have shown that genes related to the GA pathway are mainly divided into two categories: one group includes biosynthesis-related genes, such as *G2OX1*, *G2OX6*, and so on [32]. The other category includes key genes for GA signal transduction, such as *GAI* [43]. Our results showed that the binding of GA to the receptor *GID1B* mediates the ubiquitination and degradation of the growth-inhibitory factor of DELLA protein, relieving its inhibitory effect on flower induction and promoting flowering [9]. In our study, *GAI* was a negative regulator of the GA pathway, and its expression gradually decreased during the whole flower induction process. It has been reported that GA promotes the expression of *LFY* by binding the *GAMYB* protein to its promoter, and affects floral transformation by regulating *SOC1* in the SAM [44]. In addition, GA can also inhibit the expression of the flowering inhibitory gene *SVP* in the SAM, promoting the expression of the flower integrator *FT*, thereby directly inducing flowering [45]. The aging pathway can regulate flower formation independently of the photoperiodic pathway and the gibberellin pathway, as it is only related to the plant’s age and includes two important regulators, miR156 and miR172 [46]. In this study, multiple *SPLs* genes regulated by miR156 were identified and classified into two categories. *SPL9* was highly expressed in the BS2 stage and was closely related to floral transformation, while *SPL3/4* promoted the identity of the floral meristem. In addition, *SPL* can co-activate the expression of *FUL* with *SOC1* [47]. Our results showed that *FUL* was highly expressed during the BS2–BS6 stages, not only promoting the formation of inflorescence meristems, but also regulating the development of flower organs.

In summary, the hydrolysis of starch may be a signaling molecule that promotes the transition from vegetative growth to reproductive growth in *H. macrophylla* “Endless Summer”. Sucrose can promote the positive expression of *GI* and increase T6P levels, thereby participating in flower induction. ABA promotes flower induction by promoting the expression of *COL9* and participating in the photoperiodic pathway, as well as promoting the expression of *AP1*, which promotes the development of floral organs. *ARF* downregulates the synthesis and signal transduction of ZR, thus regulating the maintenance and termination of flower meristems. GA promotes the expression of *LFY* and affects the transformation of flowers by regulating *SOC1* in the SAM. The flow of auxin between sources and sinks in *H. macrophylla* is involved in the regulation of flower induction as a signal. In addition, IAA participates in the signal transduction process of GA by promoting the expression of *G2OXs* and inhibiting the expression of DELLA. As a result, multiple pathways, including the photoperiodic pathway, the aging pathway, and the gibberellin pathway, which are mainly mediated by crosstalk between sugar and hormone signals, regulate the molecular network involved in flower induction in *H. macrophylla* “Endless Summer”. The period of flower induction varies according to the physiological condition and development of the basal shoots sprouted in the spring, presenting a landscape of continuous flowering. 

## 4. Materials and Methods

### 4.1. Plant Materials

Beijing Forestry University provided the research materials used in this study. *H. macrophylla* “Endless Summer”, which had been grown in the open air grown and showed typical healthy flowering, was chosen as the test subject. The terminal buds and inflorescences were monitored and photographed every 5 to 10 days from 20 March 2021 to 30 September 2021. Fresh samples were utilized for the phenotypic observations, while frozen samples were kept in a standby ultra-low temperature refrigerator at −80 °C after being frozen using liquid nitrogen. Seven experimental materials were selected, as shown in Figure 1a,e,g,h,i,k,l, which ranged from vegetative buds (BS1) to inflorescences (BS7) at the full flowering stage.

### 4.2. Measurement of Hormone Content during Flower Development

The extraction, purification, and determination of endogenous levels of IAA, GA_3_, ABA, and ZR were performed by an indirect ELISA technique as described by He [48] and Yang et al. [49]. After being homogenized in liquid nitrogen, samples from the 7 stages (BS1-BS7) were extracted in cold 80% (*v/v*) methanol with butylated hydroxytoluene (1 mmol/L) for 5 h at 4 °C. The extracts were collected after centrifugation at 3500 r/min (4 °C) for 8 min. The supernatant was passed through Chromosep C18 columns (C18 Sep-Park Cartridge, Waters Corp., Millford, MA, USA). The hormone fractions were prewashed with 1 mL of 80% (*v/v*) methanol and eluted with 5 mL of 100% (*v/v*) methanol, 5 mL of ether, and 5 mL of 100% (*v/v*) methanol from the columns. Then they were dried under N_2_ and dissolved in 2 mL of phosphate-buffered saline (PBS) containing 0.1% (*v/v*) Tween 20 and 0.1% (*w/v*) gelatin for analysis by ELISA. The samples were diluted an appropriate amount of the standard sample with PBS to 8 concentrations (including 0 ng/mL). These series of standard samples and test samples were added to the 96-well ELISA plate, then antibodies were added and incubated at 37 °C for 30 min. After the samples had been washed four times with a PBS + Tween 20 (0.1% (*v*/*v*)) buffer, 10 mL of the diluted enzyme-linked secondary antibody was added and the samples were incubated at 37 °C for 30 min and then washed as above. Finally, the buffered enzyme substrate (orthophenylenediamino) was added, and the enzyme reaction was carried out in the dark at 37 °C for 15 min then terminated using 50 μL of 2 mol/L H_2_SO_4_. As described by Weiler et al., calculations of the enzyme immunoassay data were made [50]. The results are the means ± SE of at least three replicates.

### 4.3. Transcriptome Sequencing

Total RNA was extracted from the samples at 7 stages during flower development using the mirVana ^TM^ miRNA ISOlation Kit, Ambion-1561 (Shanghai, China). The integrity, purity, and concentration of the total RNA were determined using a gel imaging system (Tanon 2500, Shanghai, China) and an ultraviolet spectrophotometer (NanoDrop 2000, Thermo, Waltham, MA, USA). A further RNA high-throughput sequencing library was built using the qualifying samples. Shanghai Ouyi Biotechnology Co., Ltd (Shanghai, China). was the sequencing unit.

The mRNA was enriched from the total RNA by using magnetic beads. An interruption reagent was added to fragment the mRNA into smaller pieces. Short-segment mRNA was used as a template to create the first strand of cDNA using random primers. DNA polymerase I, RNase H, dNTP, and s buffer were used to create the second strand. The steps of end repair, the addition Poly A, and sequence connection were all completed after the double-stranded cDNA had been purified. The library was then created by PCR amplification after the size of the fragments had been screened. The created library was utilized for double-ended sequencing on the Illumina sequencer (Genedenovo Biotechnology, Guangzhou, China) after passing the quality assessment using the Agilent 2100 Bioanalyzer (Agilent Technologies, Santa Clara, CA, USA).

We removed the joints and carried out quality control using Trimmatic software. On this basis, we filtered out low-quality bases and N-bases, and finally obtained high-quality clean reads. The transcript was reconstructed and assembled by Trinity (version: 2.4) and CD-HIT software via de novo assembly to obtain the final unigenes.

### 4.4. Functional Annotation of Unigenes and Screening for Differentially Expressed Genes

The derived unigenes’ functions were annotated using Diamond software [51] and compared with the NR, KOG, GO, SwissProt, eggNOG, and KEGG databases. We used the HMMER program for a comparison with the Pfam database [52]. To determine the number of reads matched to the unigenes in each sample, the sequence similarity comparison technique and the Bowite2 program [53] were used, with the spliced unigene database. The value of fragments per kilobase of the exon model per million mapped fragments (FPKM) of each unigene in each sample was calculated using the eXpress program [54]. DESeq2 software was used to normalize the counts for each sample gene, according to the FPKM [55]. DEGs were selected with a |log_2_ fold change| ≥ 1 and a *p*-value of < 0.05 as the threshold by computing the multiple differences and running the significant difference test. GO and KEGG functional pathway enrichment analyses were used to screen the flowering-related DEGs. The time-series cluster method was used to analyze the expression profiles of DEGs involved in sugar, hormone, and flowering process. Pearson’s correlation coefficient (r) was used to determine the similarities among the genes (|r| ≥ 0.9) based on the FPKM of the DEGs. The visualization of the gene co-expression network was constructed using Cytascape 3.8.2 [56]. Basic parameter analysis was performed using the Network Analyzer plug-in, and cytoHubba created the co-expression network of flower induction in *H. macrophylla* “Endless Summer”. The remaining parameters had the default system settings.

### 4.5. Real-Time Quantitative Polymerase Chain Reaction

Six DEGs associated with flower development were randomly chosen for the qRT-PCR investigation to confirm the correctness of the transcriptome data. RNA was reverse-transcribed into cDNA using the HiScript III RT SuperMix for qPCR (+gDNA wiper) from Vazyme Biotech, Nanjing, China. The particular primers (Table 5) of each candidate gene were created using the program Primer Premier 5 (Premier Biosoft, San Francisco, CA, USA) and produced by Bioengineering Co., Ltd. (Shanghai, China) in accordance with the design principles of qRT-PCR primers. The reaction system was set up with TB Green^®^ Premix Ex TaqTM II (Takara, Shiga, Japan), following the manufacturers’ instructions, and PCR amplification was carried out using *RPL34* as the internal reference gene [57]. The reaction system and reaction processes are shown in Table 6. The expression levels of the DEGs were normalized and mapped by using the 2^−ΔΔCT^ method.

## Figures and Tables

**Figure 1 ijms-24-07691-f001:**
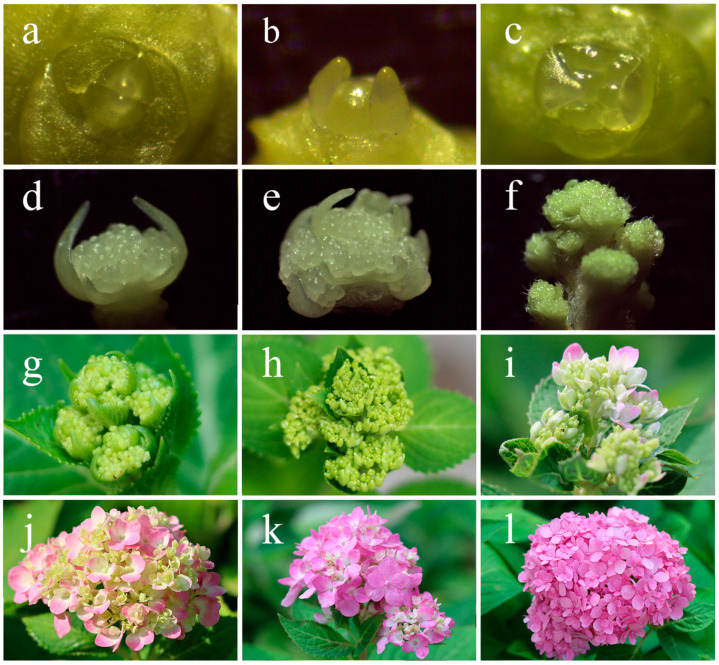
Phenotypic observations of different development stages of *H. macrophylla* “Endless Summer”. (**a**) Vegetative bud; (**b**) flower bud expansion; (**c**) meristem division; (**d**) differentiation of the floral primordia; (**e**) differentiation of floral organs; (**f**) primary inflorescence branches, (**g**) secondary inflorescence branches, (**h**) the appearance of decorative sepals, (**i**) a few colored decorative sepals, (**j**) partly colored decorative sepals; (**k**) fully colored decorative sepals; (**l**) all decorative and non-decorative flowers were colored.

**Figure 2 ijms-24-07691-f002:**
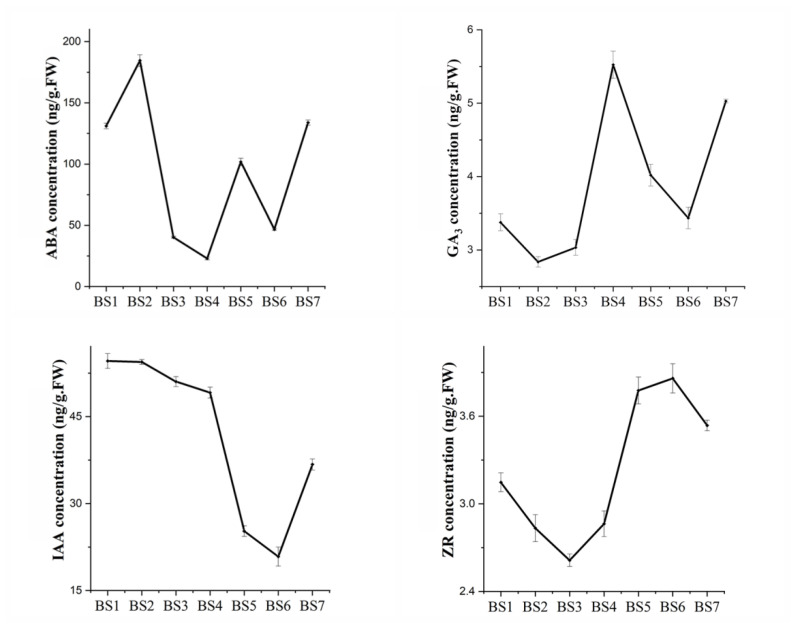
Changes in the endogenous hormones of *H. macrophylla* “Endless Summer”. The horizontal axis in the figure represents the seven periods of flower development, and the vertical axis represents changes in the hormone content at different periods. Error bars show the standard error of three biological replicates (*n* = 3).

**Figure 3 ijms-24-07691-f003:**
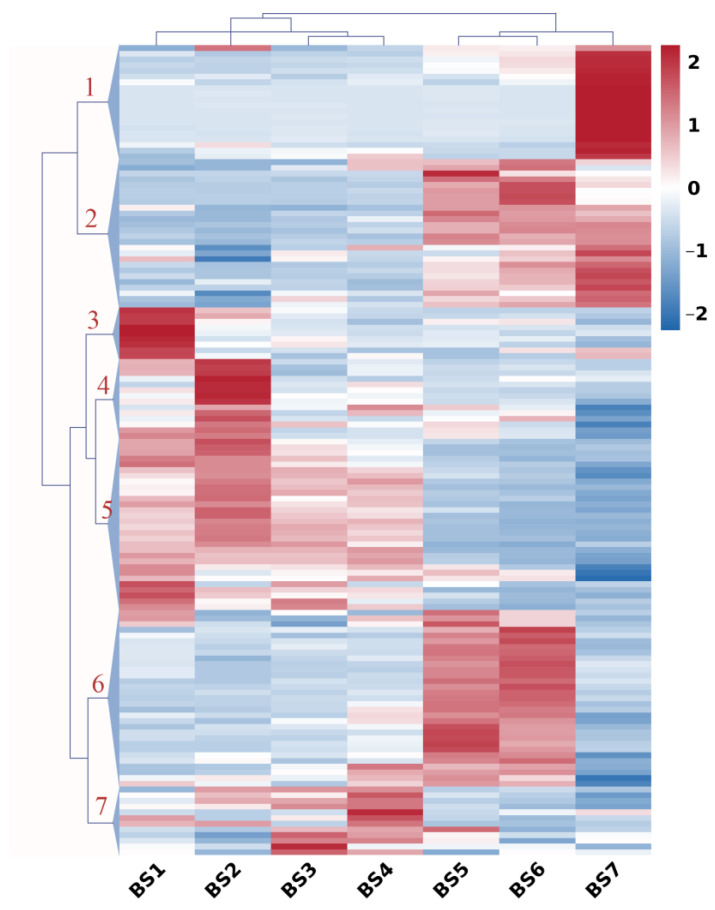
Expression profiles of sugar-related genes during flower development (BS1–BS7) in *H. macrophylla* “Endless Summer”. Hierarchical cluster analysis was performed on sugar-related genes with similar expression patterns. Fragments per kilobase of the exon model per million mapped fragments (FPKM) were used for the cluster analysis. Expression data for a given gene are shown relative to its expression during the BS1 to BS7 stages of flower development. Numbers assigned to the major clusters are indicated on the dendrogram.

**Figure 4 ijms-24-07691-f004:**
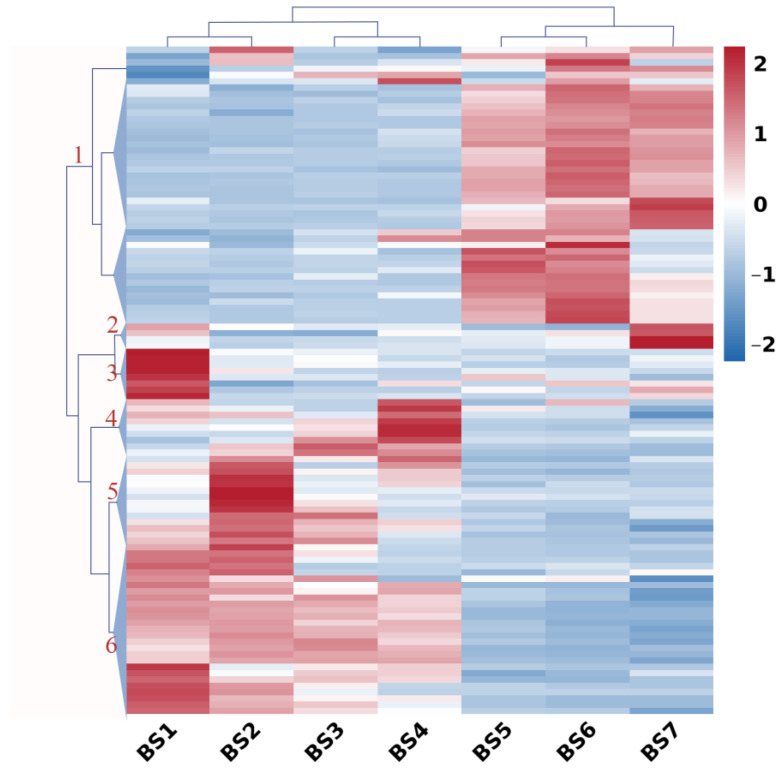
Expression profiles of hormone-related genes during flower development (BS1−BS7) in *H. macrophylla* “Endless Summer”. Hierarchical cluster analysis was performed on hormone-related genes with similar expression patterns. FPKM values were used for the cluster analysis. Expression data for a given gene are shown relative to its expression during the BS1 to BS7 stages of flower development. Numbers assigned to the major clusters are indicated on the dendrogram.

**Figure 5 ijms-24-07691-f005:**
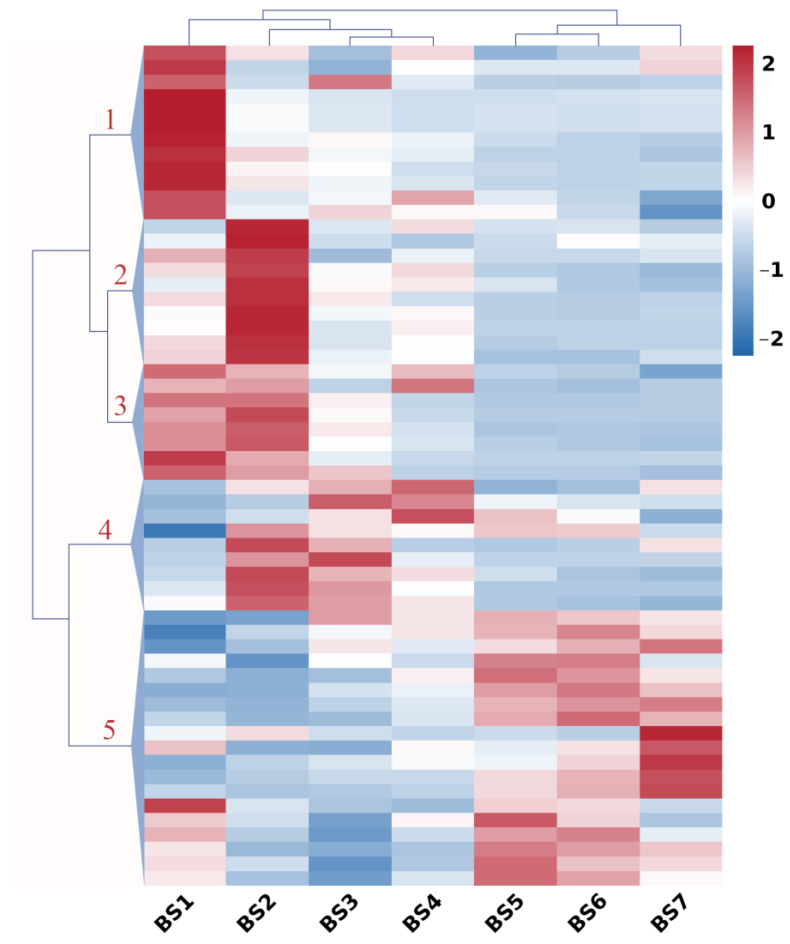
Expression profiles of flower-related genes during flower development (BS1−BS7) in *H. macrophylla* “Endless Summer”. Hierarchical cluster analysis was performed on flower-related genes with similar expression patterns. FPKM values were used for the cluster analysis. The expression data for a given gene are shown relative to its expression during the BS1 to BS7 stages of flower development. Numbers assigned to the major clusters are indicated on the dendrogram.

**Figure 6 ijms-24-07691-f006:**
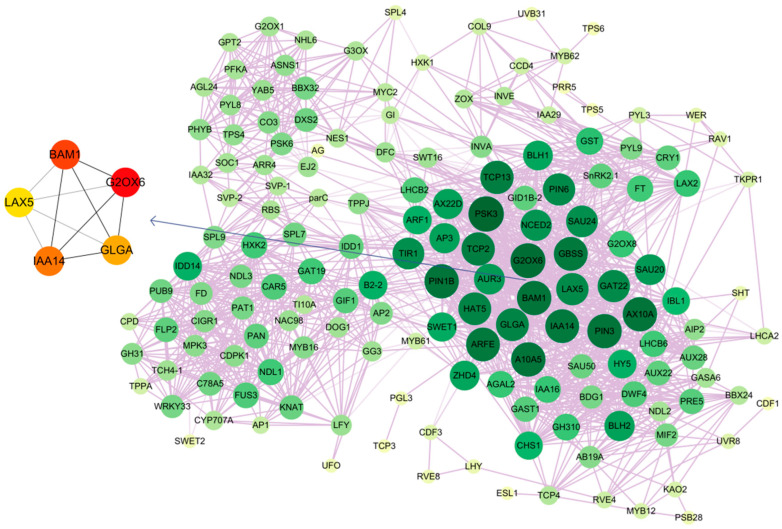
Co-expression network of DEGs related to flower development in seven stages. Colors ranging from yellow (lowest degree) to dark green (highest degree) represent the level of connectivity. The network was visualized in Cytoscape. Spheres (nodes) represent genes, and the transparency of the lines (edges) represents the strength of the correlations between genes.

**Figure 7 ijms-24-07691-f007:**
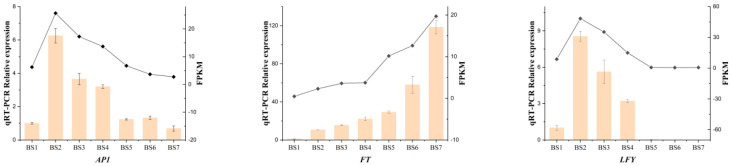
qRT−PCR validation of the expression patterns of six DEGs. The x−axis represents the seven developmental stages; the left y−axis represents the relative expression levels in the qRT−PCR results; the right y−axis represents the FPKM values from RNA−seq. Error bars show the standard error of three biological replicates (*n* = 3).

**Figure 8 ijms-24-07691-f008:**
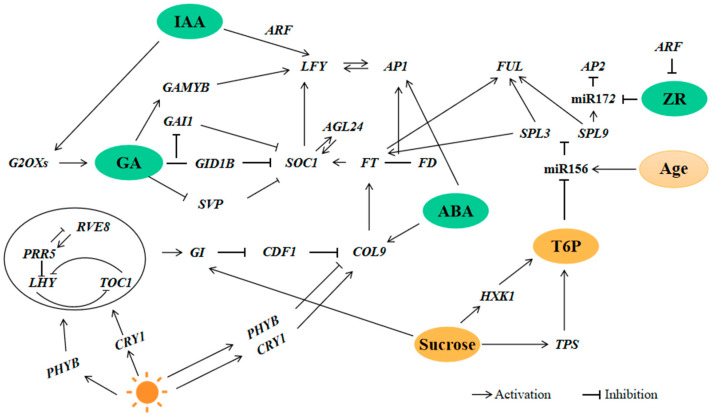
Hypothetical model of the complex gene network of the regulatory mechanism of sugars, hormones, and multiple pathways including the photoperiodic pathway, the gibberellin pathway, and the aging pathway, which mediate and regulate flower induction in *H. macrophylla* “Endless Summer”.

**Table 1 ijms-24-07691-t001:** Statistics of the quality of the sequencing data.

Sample	Raw Reads	Raw Bases	Clean Reads	Clean Bases	Valid Bases	Q30	GC
BS1	51.67 M	7.75 G	50.52 M	7.36 G	94.96%	88.66%	45.55%
BS2	47.87 M	7.18 G	46.82 M	6.84 G	95.22%	91.14%	45.52%
BS3	48.92 M	7.34 G	47.94 M	6.95 G	94.68%	91.03%	45.38%
BS4	50.73 M	7.61 G	49.60 M	7.25 G	95.34%	90.85%	45.37%
BS5	49.45 M	7.42 G	48.24 M	7.06 G	95.14%	91.86%	45.60%
BS6	48.66 M	7.30 G	47.60 M	6.95 G	95.22%	90.84%	45.57%
BS7	47.49 M	7.12 G	46.46 M	6.79 G	95.31%	91.69%	45.38%

**Table 2 ijms-24-07691-t002:** Assembly statistics of the clean reads.

Term	All	N50	Total Length	Max Length	Min Length	Average Length
Unigene	47,793	1785	56,267,846	15,827	301	1177.32

**Table 3 ijms-24-07691-t003:** KEGG pathways for the flowering of *H. macrophylla* “Endless Summer”.

Pathway Definition	Pathway	Gene Number
Glycolysis/gluconeogenesis	ko00010	165
Starch and sucrose metabolism	ko00500	159
Amino sugar and nucleotide sugar metabolism	ko00520	133
Pentose and glucuronate interconversions	ko00040	86
Citrate cycle (TCA cycle)	ko00020	69
Pentose phosphate pathway	ko00030	66
Plant hormone signal transduction	ko04075	274

**Table 4 ijms-24-07691-t004:** Flowering-related GO terms of *H. macrophylla* “Endless Summer”.

GO ID	Term	Gene Number
GO:0009908	Flower development	234
GO:0030154	Cell differentiation	183
GO:0040008	Regulation of growth	172
GO:0009555	Pollen development	146
GO:0007623	Circadian rhythm	104
GO:0009909	Regulation of flower development	104
GO:0009860	Pollen tube growth	86
GO:0009846	Pollen germination	76
GO:0048573	Photoperiodism, flowering	52
GO:0009911	Positive regulation of flower development	41
GO:0042752	Regulation of circadian rhythm	41
GO:0080092	Regulation of pollen tube growth	33
GO:0010073	Meristem maintenance	32
GO:0010584	Pollen exine formation	32
GO:0048653	Anther development	28
GO:0048443	Stamen development	25
GO:0010229	Inflorescence development	21
GO:0010228	Vegetative to reproductive phase transition of meristem	87
GO:0009910	Negative regulation of flower development	38
GO:2000028	Regulation of photoperiodism, flowering	37

**Table 5 ijms-24-07691-t005:** The primer sequences used in qRT-PCR.

Gene Name	Forward Primer Sequence (5′–3′)	Reverse Primer Sequence (5′–3′)
*RPL34*	ACCCCCGGTGGAAAGCTAGT	ACGGTTCACAGTCCTCCGGT
*AP1*	GGCAATCCAAGACCAGAAT	GGCAGCAGGAATGAAGATG
*FT*	CCTAGTGACCCGAACCTTA	GACAGTCTGACGACCCAAC
*LFY*	TGAGCAGTGCCGTGATTT	GCCTTCTTGGCATACCTG
*COL9*	CTATAAACTCGGTGGCTGAC	TGACACGACCACATCTCACT
*G2OX6*	GAACCCTACACTGAGTCTTACC	GGAGGTCGATTACAGGAAG
*SPL9*	TCTATTGTCATCTCGCTACGG	TGCTCTGCCAAGGATGTGG

**Table 6 ijms-24-07691-t006:** The reaction system and program for qRT-PCR.

The Reaction System		The Reaction Process			
TB Green Premix Ex Taq II	10 μL	Pre-denaturation	95 °C	30 s	1 cycle
Forward Primer	4 μL	PCR reaction	95 °C	5 s	40 cycles
Reverse Primer	4 μL		60 °C	30 s	
cDNA	2 μL		72 °C	30 s	
Total volume	20 μL	Melting curve	60–95 °C	15 s	

## Data Availability

All data generated or analyzed during this study are available.

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
