# Peer review of "Study on the Flower Induction Mechanism of Hydrangea macrophylla"

_ijms, 2023, doi:10.3390/ijms24097691_

Round 1
Reviewer 1 Report
Please read the review carefully

Author Response
Dear Reviewers:
Thank you for your letter and for the reviewers’ comments concerning our manuscript entitled “Study on the Flower Induction Mechanism of Hydrangea macrophylla(ijms-2287749)”. Those comments are all valuable and very helpful for revising and improving our paper as well as the important guiding significance to our researches. We have studied comments carefully and have made correction which we hope meet with approval.Revised portion are marked in red in the paper.The main corrections in the paper and the responds to the reviewer’s comments are as following:
Responding to the reviewer’s comments:
1.Response to comment: While the aim is useful, the manuscript lacks crucial information and the results are not properly described and discussed in context.
Response: We think this is an excellent and constructive suggestion. We have revised the results and discussion section. We have added cluster analysis of genes related to sugar, hormones, and flowering, and the co expression network diagram was presented in a new form.
How the paper is modified: The changes have been highlighted in line 151-464.
2.Response to comment: What does glucose metabolism mean? How can glucose metabolism contribute to the flowering process in relation to other parameters? The sentence is confused and too general.
Response: Sorry for our carelessness and poor expression. We have reanalyzed the sugar related genes and revised our conclusions.
How the paper is modified: The expression profiles of genes involved in sugar biosynthesis and metabolism showed that Beta-amylase gene BAM1 displayed a high expression level at BS2 stage, indicating that the hydrolysis of starch is a transient signal that induces the initiation of flower induction in H. macrophylla 'Endless Summer'.
3.Response to comment: Why was Arabidopsis used in this study? How is the relationship between hydrangea and Arabidopsis? Line 287: Why is Arabidopsis used as a reference and not hydrangea??
Response: Thanks for your comment. Arabidopsis and H. macrophylla are both dicotyledonous plants. As a model plant, Arabidopsis has mature research on its flowering induction mechanism, and has a short growth cycle. Subsequent gene function verification experiments are easy to carry out in Arabidopsis. However, there are few studies on the molecular level of H. macrophylla.
4.Response to comment: Hormone measurement: Why hormones have been measured with old-fashion methods and not with available published methods that cover even more hormones? Why only ZR and not CR was measured?
Response: Thank you very much for pointing out this issue. We agree with your opinion that hormones should be measured with available published methods that cover even more hormones. Unfortunately, due to the limited funding, and the difficulty in the samples collection and experiment conduction (the COVID-19 epidemic), we measured hormones with old-fashion methods. But your opinion is very helpful for us to conduct the other experiments. CR, as a new plant hormone, we will measure it in the after study. As I am facing graduation, I have no time and materials to continue doing the CR assay experiment, but our research group will definitely supplement the experiment and improve the experimental results in the future.
5.Response to comment: For all figures, the legend is useless because it does not contain any information about the diagrams. It is not a scientific legend where the reader understands what the
diagrams describe.
Response: Thank you for this comment, we have adjusted it in this revision.
How the paper is modified:
Figure 1. Phenotypic observation on different development stages of H.macrophylla 'Endless Summer'. a: vegetative bud; b: flower bud expansion; c: meristem division; d: differentiation of floral primordia; e: differentiation of floral organs; f: primary inflorescence branches, g: secondary inflorescence branches, h: the appearance of decorative sepals, i: a few colored decorative sepals, j: one-half colored decorative sepals;k: fully colored decorative sepals; l: all decorative and non decorative flowers were colored.
Figure 2. Changes of endogenous hormones of H.macrophylla 'Endless Summer'. The horizontal axis in the figure represents the seven periods of flower development, and the vertical axis represents changes in hormone content at different periods. Error bars show the standard error between three biological replicates (n = 3).
Figure 3. Expression profiles of sugar-related genes during flower development(BS1-BS7) in H.macrophylla 'Endless Summer'. Hierarchical cluster analysis was performed on sugar-related genes with similar expression patterns. FPKM values were used for the cluster analysis. Expression data for a given gene are shown relative to its expression during BS1to BS7 stages of flower development. Numbers assigned to major clusters are indicated on the dendrogram.
Figure 4. Expression profiles of hormone-related genes during flower development(BS1-BS7) in
'Endless Summer'. Hierarchical cluster analysis was performed on hormone-related genes with similar expression patterns. FPKM values were used for the cluster analysis. Expression data for a given gene are shown relative to its expression during BS1to BS7 stages of flower development. Numbers assigned to major clusters are indicated on the dendrogram.
Figure 5. Expression profiles of flower-related genes during flower development(BS1-BS7) in H.macrophylla 'Endless Summer'. Hierarchical cluster analysis was performed on flower-related genes with similar expression patterns. FPKM values were used for the cluster analysis. Expression data for a given gene are shown relative to its expression during BS1to BS7 stages of flower development. Numbers assigned to major clusters are indicated on the dendrogram.
Figure 6. Co-expression network of DEGs related to flower development in seven stages. Color range from yellow (lowest degree) to dark green (highest degree) represent connectivity level. Network was visualized in Cytoscape. Spheres (nodes) represent genes, and the transparency of the lines(edges) represents the strength of the correlation between genes.
Figure 7. qRT-PCR validation of the expression patterns of 6 DEGs. The x-axis represents the seven developmental stages; the left y-axis represents the relative expression levels of RT-qPCR results; the right y-axis represents the FPKM values from RNA-seq. Error bars show the standard error between three biological replicates (n = 3).
Figure 8. Hypothetical model of complex genes network regulatory mechanism of sugar ,hormone,and multiple pathways including the the photoperiodic pathway, the gibberellin pathway, and the age pathway mediating regulate flower induction in H.macrophylla 'Endless Summer'.
6.Response to comment: Figure 4 and Figure 10 are not mentioned or explained at all in the manuscript, which is unacceptable. Figure 3 is useless and can be omitted from the manuscript.
Response: Thanks for your comment. Figure 3 and Figure 4 was omitted from the manuscript. Figure 10(now Figure 8) was explained and mentioned at line 354,374,392,410,426.
7.Response to comment: Line 303: How was this shown (companion cells!), any experimental evidences?
Response: Thanks for your comment. We referred to the study of Arabidopsis, as described below: The overall picture is that CO acts in the phloem companion cells of leaves and that its main effect is to induce FT mRNA in these cells [11, 12, 14–19]. Surprisingly, FT, a small globular protein of 20 kDa, interacts at the shoot apex with the bZIP transcription factor FLOWERING LOCUS D (FD) to induce downstream targets. Given that green fluorescent protein (GFP), which as a monomer is 27 kDa, can be easily exported to sink tissue including flowers when expressed in phloem companion cells, the latter finding strongly implied that FT protein is the mobile floral-inductive signal [17–19]. In agreement with this hypothesis, an FT-GFP fusion, just like GFP, can be exported from the phloem of both rice and Arabidopsis.
Mathieu J , Warthmann N , F Küttner, et al. Export of FT protein from phloem companion cells is sufficient for floral induction in Arabidopsis[J]. Current Biology, 2007, 17(12):1055-1060.
DOI:10.1016/j.cub.2007.05.009
- Response to comment: Lines 340-350: Glucose and sucrose are confusedly discussed! Glc and sucrose measurements? Trehalose6P is involved in starch metabolism and not in sucrose metabolism?
Response: Sorry for the careless discussion about glucose and sucrose. We have revised it at line .As for T6P involved in starch metabolism or sucrose metabolism, here is the reference: Feeding of trehalose to Arabidopsis leaves led to stimulation of starch synthesis within 30 min, accompanied by activation of ADP-glucose pyrophosphorylase (AGPase) via posttranslational redox modification. We also analyzed transgenic Arabidopsis plants with T6P levels increased by expression of T6P synthase. Compared with wild type, leaves of T6P synthase-expressing plants had increased redox activation of AGPase and increased starch, whereas TPP-expressing plants showed the opposite. Incubation of intact isolated chloroplasts with 100 {micro}M T6P significantly and specifically increased reductive activation of AGPase within 15 min. Results provide evidence that T6P is synthesized in the cytosol and acts on plastidial metabolism by promoting thioredoxin-mediated redox transfer to AGPase in response to cytosolic sugar levels, thereby allowing starch synthesis to be regulated independently of light.
Kolbe A , Tiessen A , Schluepmann H , et al. Trehalose 6-phosphate regulates starch synthesis via posttranslational redox activation of ADP-glucose pyrophosphorylase[J]. Proceedings of the National Academy of Sciences of the United States of America, 2005.
DOI:10.1073/pnas.0503410102
9.Response to comment: How do you know that sucrose promoted the production of T6P, any measurement, any external application?
Response: Thanks for your comment. As for the discription that sucrose promoted the production of T6P, here is our reference:
Further, to better understand the coordination between sugars, trehalose pathway, and temperature-dependent growth, we examine the interrelationship between sugars, trehalose phosphate synthase (TPS), and trehalose phosphate phosphatase (TPP) gene expression and T6P content in seedlings. Sucrose, particularly when fed exogenously, correlated well with TPS1 and TPPB gene expression, suggesting that these enzymes are involved in maintaining carbon flux through the pathway in relation to sucrose supply.
Nunes C , Schluepmann H , Delatte T L , et al. Regulation of growth by the trehalose pathway: relationship to temperature and sucrose.[J]. Plant Signaling & Behavior, 2013, 8(12):e26626.
DOI:10.4161/psb.26626
10.Response to comment: Lines 350-360: repetition of the previous chapter! Useless.
Response: Thank you for your comment. The repetition section was omitted.
11.Response to comment: Conclusions are completely misplaced, should be at the end of discussion.
Response: Thanks for your comment. We have changed the position of the Conclusion in line 460-465.
Reviewer 2 Report
In this study, the authors conducted ELISA and transcriptome sequencing to identify differentially expressed genes to explore the flower induction of Hydrangea macrophylla. They showed that a high concentration of auxin, ABA, and low concentration of gibberellin and zeatin nucleoside promote flowering. A total of 158 DEGs involved in flower induction were identified through the enrichment of DEGs and gene co-expression network analysis. The results provide helpful information of the molecular mechanism of continuous flowering in H. macrophylla. However, there is no functional test in plants in for the key genes identified from this study, such as over-expression of 1-2 hub genes in Arabidopsis. I suggest acceptance of the manuscript after the major reversion.
Author Response
Dear Reviewers:
Thank you for your letter and for the reviewers’ comments concerning our manuscript entitled “Study on the Flower Induction Mechanism of Hydrangea macrophylla(ijms-2287749)”. Those comments are all valuable and very helpful for revising and improving our paper as well as the important guiding significance to our researches. We have studied comments carefully and have made correction which we hope meet with approval.Revised portion are marked in red in the paper.The main corrections in the paper and the responds to the reviewer’s comments are as following:
Responding to the reviewer’s comments:
1.Response to comment: There is no functional test in plants in for the key genes identified from this study, such as over-expression of 1-2 hub genes in Arabidopsis.
Response: Thank you very much for pointing out this significant issue. However, our research mainly focuses on the molecular level of flower induction in Hydrangea macrophylla, and it is also complete without conducting functional verification experiments. We do agree with your opinion that we should over-express 1-2 hub genes in Arabidopsis to verify the gene function. Unfortunately, due to the difficulty in the samples collection and experiment conduction (the COVID-19 epidemic), we didn't carry out this experiment before. Your opinions are very helpful for us to continue our research, even though I'm about to graduate, our research group will definitely supplement the experiment and improve the experimental results in the future.
How the paper is modified: We we have revised the manuscript in Results, Discussion and Abstract.
Round 2
Reviewer 1 Report
Dear authors, the revised manuscript has been improved, but there are still some parts that need to described more clearly and distictly, especially the methods and discussion sections. There are also many spelling errors in the new parts and the citation of the figures is missing in the text.

Author Response
Dear Reviewers:
Thank you for your letter and for the reviewers’ comments concerning our manuscript entitled “Study on the Flower Induction Mechanism of Hydrangea macrophylla(ijms-2287749)”. Those comments are all valuable and very helpful for revising and improving our paper as well as the important guiding significance to our researches. We have studied comments carefully and have made correction which we hope meet with approval.Revised portion are marked in red in the paper.The main corrections in the paper and the responds to the reviewer’s comments are as following:

Reviewer 2 Report
The authors have made several changes to address the issues raised in preview review. I am largely happy with the changes they have made. I suggest acceptance of the revised manuscript.
Author Response

(The authors gave the same response as above.)
